# FreeLong: Training-Free Long Video Generation with SpectralBlend Temporal Attention

**Yu Lu[1], Yuanzhi Liang[2], Linchao Zhu[1], Yi Yang[1]** *

[1]The State Key Lab of Brain-Machine Intelligence, Zhejiang University
[2]University of Technology Sydney

## Abstract

Video diffusion models have made substantial progress in various video generation applications. However, training models for long video generation tasks require significant computational and data resources, posing a challenge to developing long video diffusion models. This paper investigates a straightforward and training-free approach to extend an existing short video diffusion model (*e.g.*, pre-trained on 16-frame videos) for consistent long video generation (*e.g.*, 128 frames). Our preliminary observation has found that directly applying the short video diffusion model to generate long videos can lead to severe video quality degradation. Further investigation reveals that this degradation is primarily due to the distortion of high-frequency components in long videos, characterized by a decrease in spatial high-frequency components and an increase in temporal high-frequency components. Motivated by this, we propose a novel solution named FreeLong to balance the frequency distribution of long video features during the denoising process. FreeLong blends the low-frequency components of global video features, which encapsulate the entire video sequence, with the high-frequency components of local video features that focus on shorter subsequences of frames. This approach maintains global consistency while incorporating diverse and high-quality spatiotemporal details from local videos, enhancing both the consistency and fidelity of long video generation. We evaluated FreeLong on multiple base video diffusion models and observed significant improvements. Additionally, our method supports coherent multi-prompt generation, ensuring both visual coherence and seamless transitions between scenes. *Our project page is at: https://yulu.net.cn/freelong*.

## 1 Introduction

Video diffusion models [1, 2, 3, 4, 5, 6, 7] trained on vast video-text datasets [8, 9] have demonstrated impressive capabilities in generating high-quality videos. Inspired by Sora [10], multiple studies [11, 12, 13] have concentrated on training these models to create longer videos using extensive, long video-text datasets [14, 15, 16, 17, 18]. However, these methods demand significant computational resources and data annotations.

A more practical approach involves adapting pre-trained short video models to generate consistent longer video sequences without retraining. Recent research [19, 20] has explored sliding window temporal attention to ensure smooth transitions between video clips in the generation of long videos. Nonetheless, these techniques often struggle to maintain global temporal consistency across extended sequences and require multiple passes of temporal attention.

In this study, we propose a simple, training-free method to adapt existing short video diffusion models (e.g., pretrained on 16 frames) for generating consistent long videos (e.g., 128 frames). Initially,

---

*Corresponding author

38th Conference on Neural Information Processing Systems (NeurIPS 2024)

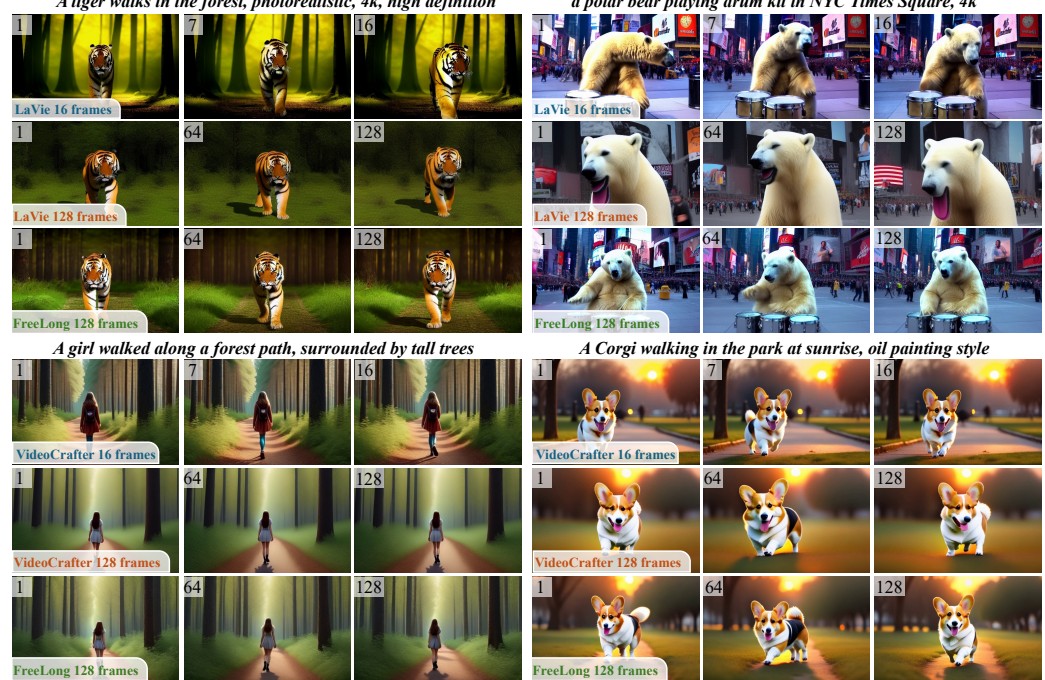

Figure 1: **Results of Short and Long Videos.** The first row of each case shows 16-frame videos generated using short video diffusion models (LaVie [1] and VideoCrafter2 [2]). Directly extending these models to longer videos, like those with 128 frames, preserves temporal consistency but lacks fine spatial-temporal details. In contrast, our proposed FreeLong adapts short video diffusion models to create consistent long videos with high fidelity.

we examine the direct application of short video diffusion models for long video generation. As depicted in Figure 1, straightforwardly using a 16-frame video diffusion model to produce 128-frame sequences yields globally consistent yet low-quality results.

To delve further into these issues, we conducted a frequency analysis of the generated long videos. As shown in Figure 2 (a), the low-frequency components remain stable as the video length increases, while the high-frequency components exhibit a noticeable decline, leading to a drop in video quality. The findings indicate that although the overall temporal structure is preserved, fine-grained details suffer notably in longer sequences. Specifically, there is a decrease in high-frequency spatial components (Figure 2 (b)) and an increase in high-frequency temporal components (Figure 2 (c)). This high-frequency distortion poses a challenge in maintaining high fidelity over extended sequences. As illustrated in the middle row of each case in Figure 1, intricate textures like forest paths or sunrises become blurred and less defined, while temporal flickering and sudden changes disrupt the video's narrative flow.

To tackle these challenges, we introduce FreeLong, a novel framework that employs SpectralBlend Temporal Attention (SpectralBlend-TA) to balance the frequency distribution of long video features in the denoising process. SpectralBlend-TA integrates global and local features via two parallel streams, enhancing the fidelity and consistency of long video generation. The global stream deals with the entire video sequence, capturing extensive dependencies and themes for narrative continuity. Meanwhile, the local stream focuses on shorter frame subsequences to retain fine details and smooth transitions, preserving high-frequency spatial and temporal information. SpectralBlend-TA combines global and local video features in the frequency domain, improving both consistency and fidelity by blending low-frequency global components with high-frequency local components. Our method is entirely training-free and allows for the easy integration of FreeLong into existing video diffusion models by adjusting the original temporal attention of video diffusion models. Comparative

experiments demonstrate significant improvements in temporal consistency and video fidelity when applying our method to generate long video sequences.

Our contributions can be summarized as follows: **1)** We conduct a frequency analysis on the direct application of short video models for longer video generation and identify high-frequency distortions in the longer videos. **2)** We devise a SpectralBlend Temporal Attention mechanism to merge the consistent low-frequency components of global videos with the high-fidelity high-frequency components of local videos. **3)** Our training-free approach, FreeLong, outperforms existing state-of-the-art models in both temporal consistency and video fidelity.

## 2    Related Work

**Text-to-Video Diffusion Models:** Text-to-video (T2V) generation has progressed significantly from early variational autoencoders [21, 22] and GANs [23] to advanced diffusion-based techniques [3, 4, 24, 25, 26], marking a major leap in synthesis methods. Modern video diffusion models build on pre-trained image-to-text diffusion models [27, 28, 29], incorporating temporal transformers in the diffusion UNet to capture temporal relationships. These models achieve impressive video generation results through post-training on video-text data [14, 9, 8, 1], enhancing coherence and fidelity. However, due to computational constraints and limited dataset availability, current video diffusion models are typically trained on fixed-length short videos (*e.g.*, 16 frames), limiting their ability to produce longer videos. In this paper, we propose extending these short video diffusion models to generate long and consistent videos without requiring any additional training videos.

**Long-video Generation:** Generating long videos is challenging due to temporal complexity, resource constraints, and the need for content consistency. Recent advancements focus on improving temporal coherence and visual quality using GAN-based [30, 31] and diffusion-based techniques [32, 33, 34, 35, 36]. For instance, Nuwa-XL [36] employs a parallel diffusion process, while StreamingT2V [11] uses an autoregressive approach with a short-long memory block to improve the consistency of long video sequences. Despite their effectiveness, these methods require substantial computational resources and large-scale datasets. Recent research has explored training-free adaptations using short video diffusion models for long video generation. Gen-L-Video [37] extends videos by merging overlapping sub-segments with a sliding-window method during denoising. FreeNoise [19] employs sliding-window temporal attention and a noise initialization strategy to maintain temporal consistency. However, these approaches focus on smooth transitions between video clips and fail to capture global consistency across long video sequences. This paper proposes FreeLong, a novel approach that blends global and local video features during the denoising process to enhance both global temporal consistency and visual quality in long video generation.

## 3    Observation and Analysis

When attempting to adapt short video diffusion models to generate long videos, a straightforward approach is to input a longer noise sequence into the short video models. The temporal transformer layers in the video diffusion model are not constrained by input length, making this method seemingly viable. However, our empirical study reveals significant challenges, as demonstrated in Figure 1. Generated long videos often exhibit fewer detailed textures, such as blurred forests in the background, and more irregular variations, like abrupt changes in motion. We attribute these issues to two main factors: the limitations of the temporal attention mechanism and the distortion of high-frequency components.

**Attention Mechanism Analysis:** The temporal attention mechanism in video diffusion models is pre-trained on fixed-length videos, which complicates its ability to generate longer videos. As shown in Figure 3 , increasing video length hinders the temporal attention's ability to accurately capture frame-to-frame relationships. For 16-frame videos, the attention maps show a diagonal pattern, indicating high correlations with adjacent frames that preserve spatial-temporal details and motion patterns. In contrast, for 128-frame videos, the less structured attention maps suggest difficulty in focusing on relevant information across distant frames, leading to missed subtle motion patterns and over-smoothed or blurred generations.

**Frequency Analysis:** To better understand the generation process of long videos, we analyzed the frequency components in videos of varying lengths using the Signal-to-Noise Ratio (SNR) as a

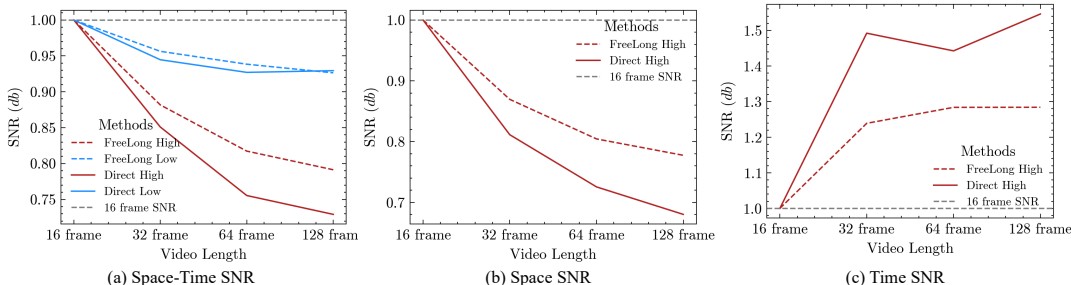

(a) Space-Time SNR  (b) Space SNR  (c) Time SNR

Figure 2: **Ratio of short video SNR on high/low frequency to different long videos.** Our findings reveal that: (a) When direct extend short video diffusion model to generate long videos, the SNR of high-frequency components in the space-time frequency domain degrades significantly as video length increases. (b) In the spatial frequency domain, the SNR of high-frequency components decreases even more substantially, resulting in the over-smoothing of each frame. (c) Conversely, in the temporal frequency domain, the SNR of high-frequency components increases significantly, introducing temporal flickering.

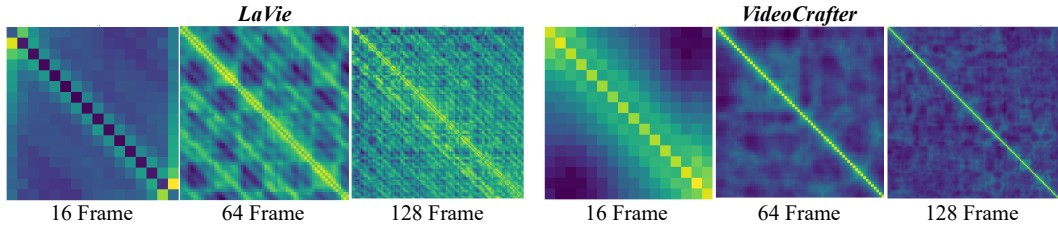

Figure 3: **Temporal Attention Visualization.** We visualize the temporal attention by average across all layers and time steps from LaVie [1] and VideoCrafter [2]. The attention maps for 16-frame videos exhibit a diagonal-like pattern, indicating a high correlation with adjacent frames, which helps preserve high-frequency details and motion patterns when generating new frames. In contrast, attention maps for longer videos are less structured, such as 128 frames, making the model struggle to identify and attend to the relevant information across distant frames. This lack of structure in the attention maps results in the distortion of high-frequency components of long videos, which results in the degradation of fine spatial-temporal details.

metric. Ideally, short video diffusion models generate 16-frame videos with high quality, and robust longer videos derived from such models should exhibit consistent SNR values across all frequency components. However, Figure 2 reveals significant differences in the SNR of high/low frequency components[2] between generated short and long videos. The SNR of low-frequency components remains relatively consistent for long videos (1.0 for 16 frames to 0.93 for 128 frames), suggesting that the model maintains overall structure and low-frequency details in extended sequences. However, the SNR of high-frequency components drops significantly for longer videos (1.0 for 16 frames to 0.73 for 128 frames), indicating a loss of fine details and increased distortion, leading to suboptimal visual fidelity.

Further investigation into the spatial and temporal frequency domains revealed two key findings: (1) In the spatial domain, the high-frequency components of long videos degrade significantly (0.68 for 128 frames), causing substantial degradation of spatial details in each frame and resulting in blurred frames. (2) In the temporal domain, the high-frequency components increase with video length (1.5 for 128-frame videos), resulting in temporal flickering and incoherent video outputs.

---

[2]We split the frequency components into high-frequency ($\phi \sim (0.25\pi - 1.00\pi)$) and low-frequency ($\phi \sim (0.00\pi - 0.25\pi)$) and compared the SNR of each component in long videos to the corresponding SNR in 16-frame videos.

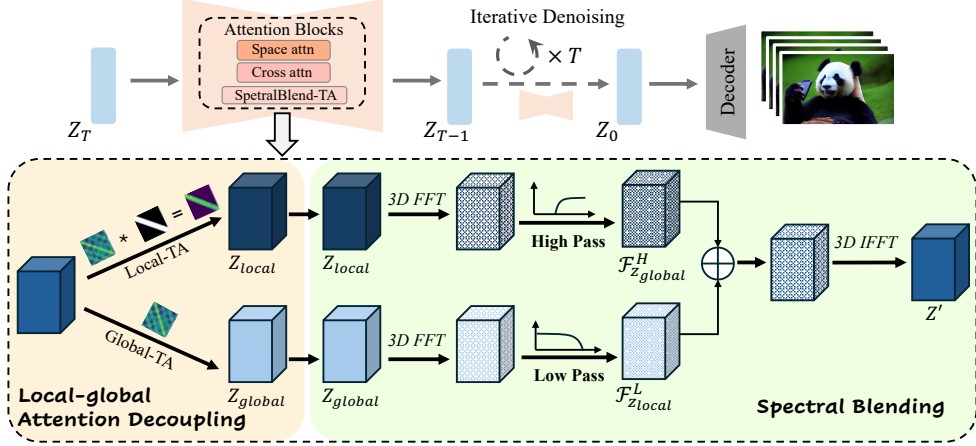

Figure 4: **Overview of FreeLong.** FreeLong facilitates consistent and high-fidelity video generation using SpectralBlend Temporal Attention (SpectralBlend-TA). SpectralBlend-TA effectively blends low-frequency global video features with high-frequency local video features through a two-step process: local-global attention decoupling and spectral blending. Local video features are obtained by masking temporal attention to concentrate on fixed-length adjacent frames, while global temporal attention encompasses all frames. During spectral blending, 3D FFT projects features into the frequency domain, where high-frequency local components and low-frequency global components are merged. The resulting blended feature, transformed back to the time domain via IFFT, is then utilized in the subsequent block for refined video generation.

## 4   FreeLong: Training-free Long Video Generation

Motivated by the above analysis, we propose FreeLong, a method designed to generate high-fidelity and consistent long videos using the inherent power of the diffusion model. As illustrated in Figure 4, our FreeLong uses a diffusion UNet from pre-trained short video diffusion models and introduces a SpectralBlend Temporal Attention (SpectralBlend-TA) to facilitate long video generation. The SpectralBlend-TA consists of two steps: local-global attention decoupling and spectral blending.

**Local-global Attention Decoupling:**

The temporal attention in short video models is optimized to model short frame sequences accurately, maintaining high-fidelity visual information. Conversely, the long-range temporal attention from short video models tends to maintain overall layout and and object consistency. Given these properties, we first decouple the local and global attention. The local attention matrix can be obtained as:

$$A_{\text{local}}(i,j) = \begin{cases} \text{Softmax}\left(\frac{Q_i K_j^\top}{\sqrt{d}}\right) & \text{if } |i-j| \le \alpha \\ 0 & \text{otherwise}, \end{cases} \tag{1}$$

where $Q$ and $K$ are the query and key matrices derived from the input video feature $Z_{in}$. The local attention $A_{\text{local}}$ leads to each frame $i$ only attending to frames within a window of $2\alpha$ frames. Given the local attention matrix $A_{\text{local}}$, the local video features $Z_{\text{local}}$ can be obtained by: $Z_{\text{local}} = A_{\text{local}}V$, where $V$ is the value matrix derived from the input video feature $Z_{in}$. By restricting the temporal attention to adjacent local frames, we preserve the capabilities of short video models, thereby retaining high-fidelity visual details in local video features.

We then define the global attention matrix where each frame attends to all other frames. The global attention matrix can be computed as follows:

$$A_{\text{global}}(i,j) = \text{Softmax}\left(\frac{Q_i K_j^\top}{\sqrt{d}}\right), \tag{2}$$

Given the global attention matrix $A_{\text{global}}$, the global video features $Z_{\text{global}}$ can be obtained by: $Z_{\text{global}} = A_{\text{global}}V$. The global video features process the entire video sequence, ensuring narrative continuity and coherence, while capturing long-range dependencies and overarching themes.

**Spectral Blending:** After obtaining the global and local video features, a frequency filter is used to blend the low-frequency components of the global video latent $Z_{global}$ with the high-frequency components of the local video latent $Z_{local}$, resulting in a new video latent $Z'$. This fused latent retains the global coherence and structure provided by $Z_{global}$, while benefiting from the enhanced high-frequency details introduced by $Z_{local}$. The process is described by:

$$\mathcal{F}^L_{z_{global}} = \text{FFT}_{3D}(Z_{global}) \odot \mathcal{P}, \tag{3}$$

$$\mathcal{F}^H_{z_{local}} = \text{FFT}_{3D}(Z_{local}) \odot (1 - \mathcal{P}), \tag{4}$$

$$Z' = \text{IFFT}_{3D}(\mathcal{F}^L_{z_{global}} + \mathcal{F}^H_{z_{local}}) \tag{5}$$

where $\text{FFT}_{3D}$ is the Fast Fourier Transformation operated on both spatial and temporal dimensions, $\text{IFFT}_{3D}$ is the Inverse Fast Fourier Transformation that maps back the blended representation $Z'$ from the frequency domain, and $\mathcal{P} \in \mathbb{R}^{4 \times N \times h \times w}$ is the spatial-temporal Low Pass Filter (LPF), which is a tensor of the same shape as the latent. The final fused video feature $Z'$ serves as the input to our subsequent video generation module.

The rationale behind using low-frequency components from the global video features and high-frequency components from the local video features stems from our analysis. The global features provide a stable, coherent structure, preserving the overall layout and object consistency throughout the video. This is crucial for maintaining temporal consistency in long videos. On the other hand, local features retain high-fidelity details, which are essential for capturing fine textures and intricate motion patterns that tend to degrade in long sequences. By blending these components in the frequency domain, we harness the strengths of both global consistency and local detail preservation, addressing the issues of blurred frames and temporal flickering observed in our analysis.

Recent studies [38, 39] indicate that latent diffusion models [27] generate varying levels of visual content at different stages of the denoising process: scene layout and object shapes in the early steps, and fine details in the later steps. We propose fusing global and local video features in the early $\tau$ steps of the denoising process and using local video features in the remaining steps. This fusion ensures that the overall layout and object appearance of the generated long video follow the global features, thereby maintaining temporal consistency in the generated videos.

## 5 Experiments

### 5.1 Implementation Details

**Baseline Models:** To evaluate the effectiveness and generalization of our proposed method, we apply FreeLong on two publicly available diffusion-based text-to-video models, LaVie [1] and VideoCrafter [2]. These models are trained on short videos with fixed length (*i.e.*, 16 frames), we extend them to produce long videos (*i.e.*, 128 frames [40]). We set $\alpha = 8$ for the local attention setting and set $\tau$ to 25. During inference, the parameters of the frequency filter for each model are kept the same for a fair comparison. Specifically, we use a Gaussian Low Pass Filter (GLPF) $\mathcal{P}_G$ with a normalized spatiotemporal stop frequency of $D_0 = 0.25$. Multi-prompt videos are generated with random noise, and FreeNoise [19] is used for single-prompt long video generation.

**Test Prompts:** We chose 200 prompts from VBench [41] to validate the effectiveness of our method.

**Evaluation Metrics:** For text-to-video generation, we employed several metrics from VBench [41] to evaluate two aspects: video consistency and video fidelity. For video consistency measurement, we use two metrics: 1). Subject consistency, computed by the DINO [42] feature similarity across frames to assess whether object appearance remains consistent throughout the whole video. 2). Background consistency, calculated by CLIP [43] feature similarity across frames. For video fidelity measurement, we use three metrics: 1). Motion smoothness, which utilizes the motion priors in the video frame interpolation model AMT [44] to evaluate the smoothness of generated motions. 2). Temporal flickering, which takes static frames and computes the mean absolute difference across frames. 3). Imaging quality, calculated using the MUSIQ [45] image quality predictor trained on the SPAQ [46] dataset.

Table 1: **Quantitative Comparison**. "Direct sampling" and "Sliding window" indicate directly sampling 128 frames and applying temporal sliding windows based on short video generation models, respectively. Compared to these methods, our FreeLong achieves consistent long video generation with high fidelity.

| Methods | Sub (↑) | Back (↑) | Motion (↑) | Flicker (↑) | Imaging (↑) | Inference Time (↓) |
|---|---|---|---|---|---|---|
| Direct sampling | 88.95 | 93.23 | 92.77 | 91.44 | 64.76 | 1.8s |
| Sliding window | 85.80 | 92.83 | 95.79 | 94.00 | 66.57 | 2.6s |
| FreeNoise [19] | 92.30 | 95.87 | 96.32 | 94.94 | 67.14 | 2.6s |
| *Ours* | **95.16** | **96.80** | **96.85** | **96.04** | **67.55** | **2.2s** |

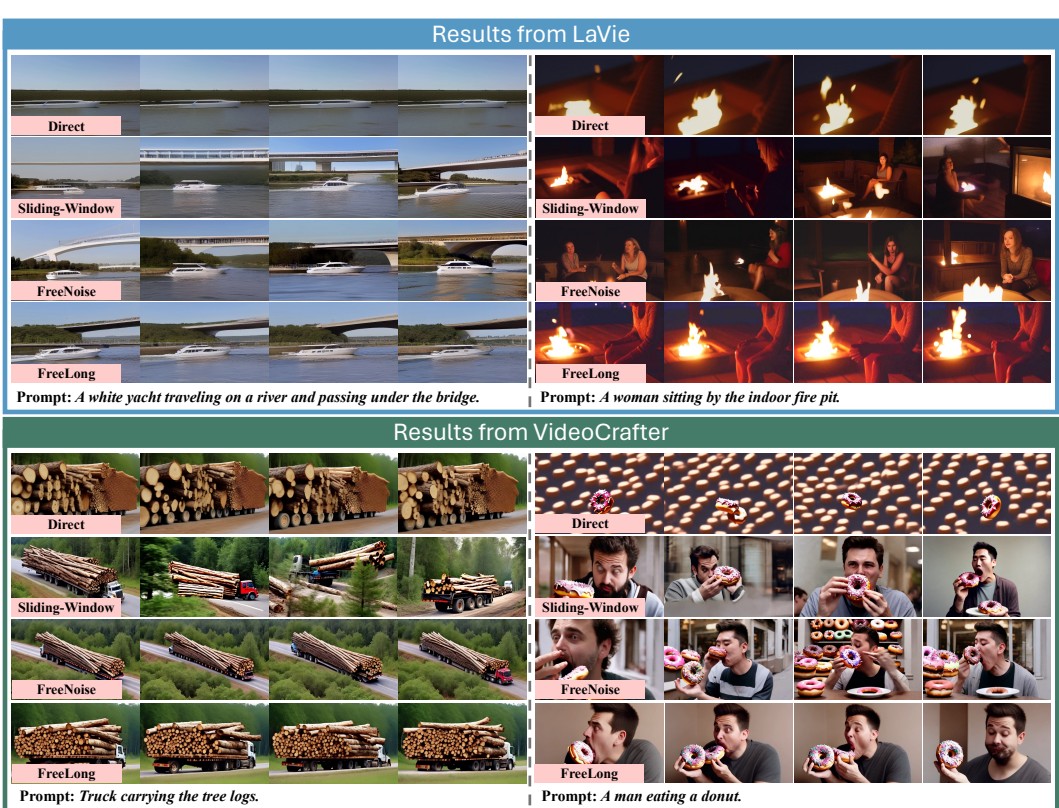

Figure 5: **Qualitative Comparison.** Results from LaVie [1] and VideoCrafter [2] are presented. Direct videos exhibit consistent frames, but they appear over-smoothed. FreeNoise and the sliding-window approach struggle to capture global consistency effectively. Our FreeLong method achieves consistent long video generation while maintaining high fidelity, preserving crucial details and textures across the entire sequence.

## 5.2 Quantitative Comparison

We compare our FreeLong method with other training-free approaches for long video generation using diffusion models. Our comparison includes three methods: (1) Direct sampling. It directly samples 128 frames from the short video models. (2) Sliding window. It adopts temporal sliding windows [20] to process a fixed number of frames at a time. (3) FreeNoise [19]. FreeNoise introduces repeat input noise to maintain temporal coherence across long sequences.

Table 1 presents the quantitative results. Direct generation of long videos suffers from high-frequency distortion, leading to significant quality degradation. This method results in low fidelity scores, including imaging quality, temporal flickering, and motion smoothness. The sliding-window method and FreeNoise show improved video quality thanks to the fixed effective temporal attention window

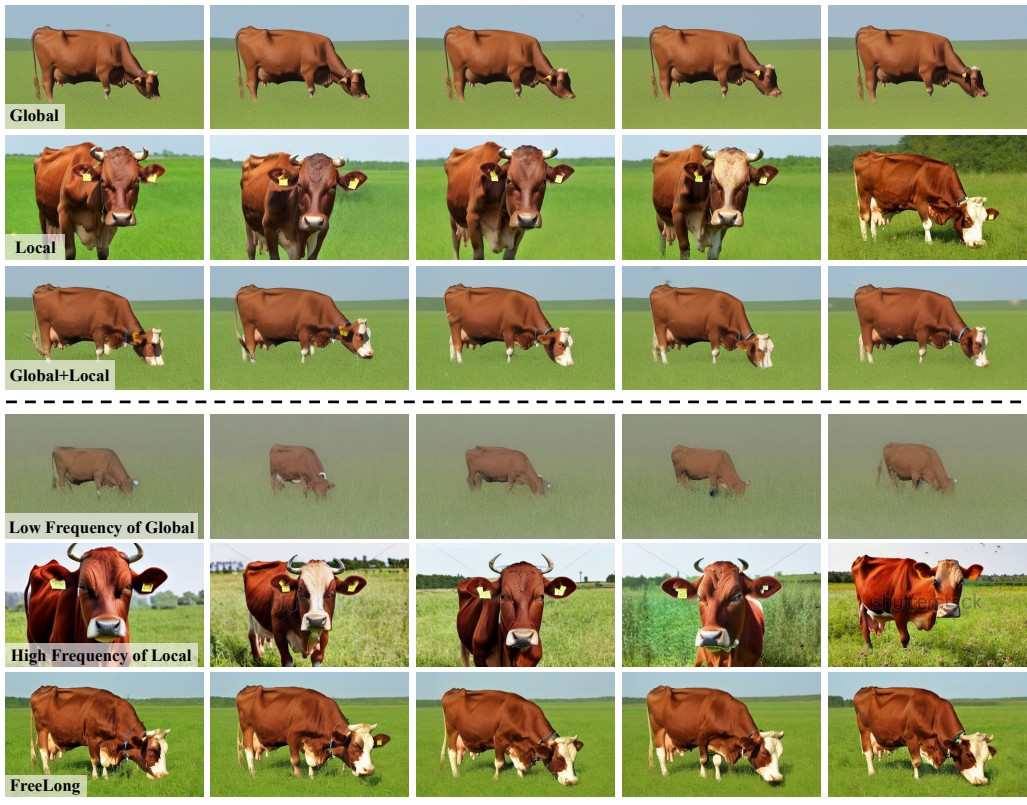

Figure 6: **Ablation Study.** Global features and low-frequency components of global features ensure consistency but degrade fidelity. Local features and high-frequency local features maintain spatial-temporal details but lack temporal consistency. Directly adding global and local features degrades fidelity. Our method achieves both high fidelity and temporal consistency.

but still face challenges in maintaining consistency across long videos. Our FreeLong method achieves the highest scores across all metrics, producing consistent long videos with high fidelity. Moreover, we also examine the inference time of these methods on the NVIDIA A100. As delineated in Table 1, our approach achieves a faster speed compared to preceding methods by employing single-pass temporal attentions.

## 5.3 Qualitative Comparison

The synthesis results of each method are shown in Figure 5. In the first row, directly sampling 128 frames through a model trained on 16 frames will bring poor quality results due to the high-frequency distortion. For example, the yacht (left) and the girl (right) have blurred and the background is not clear. As shown in second row in Figure 5, using temporal sliding windows helps generate more vivid videos, but this approach ignores long-range visual consistency, causing the subject and background to appear significantly different across frames. FreeNoise attempts to promote global consistency by repeating and shuffling the initial noise for each frame; however, it fails to maintain long-range visual consistency and suffers from content mutations. In contrast, our method, FreeLong, explicitly enforces global constraints during the denoising process, achieving temporal consistency while preserving high fidelity across frames. Results shown in Figure 5 demonstrate that FreeLong successfully renders temporal consistent longer videos, outperforming all other methods.

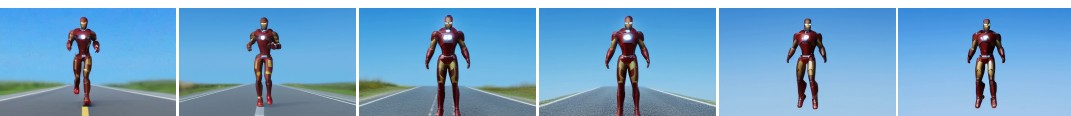

Prompt1: Ironman running on the road, 4K, high resolution
Prompt2: Ironman standing on the road, 4K, high resolution
Prompt3: Ironman flying on the sky, 4K, high resolution

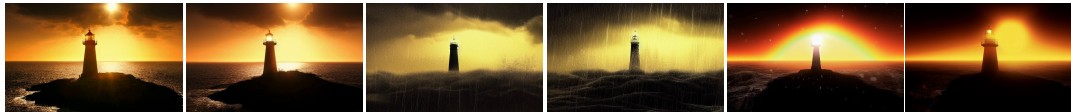

Prompt1: The morning sun rises, illuminating a solitary lighthouse on a rocky shore.
Prompt2: Clouds gather, and rain starts to fall, the lighthouse stand firm under storm with heavy rain and thunder.
Prompt3: The storm clears, the lighthouse stand under rainbow.
Prompt4: Night falls, and the lighthouse raise light through the darkness.

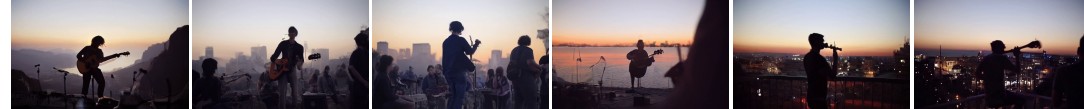

Prompt1: A musician with a guitar performs at the edge of a mountain at morning.;
Prompt2: The musician performs in a busy urban park, a crowd gathers, enchanted by the music.;
Prompt3: A serene lakeside at sunset where the musician plays alone, reflecting.;
Prompt4: Night falls, and the musician joins a lively street festival, lights and music filling the air.;
Prompt4: The journey ends on a quiet balcony overlooking the city at night, the musician performs towards the city.

Figure 7: **Results of Multi-Prompt Video Generation.** Our method ensures coherent visual continuity and motion consistency across different video segments.

## 5.4 Ablation Studies

To validate the effectiveness of each module in our FreeLong method—global video feature, local video feature, and our combined approach—we present the generated results by ablating each component.

As shown in the top part of Figure 6, videos generated solely from global video features maintain consistent content but suffer from severe fidelity degradation. Conversely, videos generated using only local video features preserve fidelity due to the fixed effective temporal attention window but fail to maintain temporal consistency, as evidenced by the changing color of the cow. Simply combining global and local video features results in fidelity degradation because the high-frequency components of the global video features degrade significantly.

In the bottom part of Figure 6, we show the videos generated by combining the low-frequency components from global video features with the high-frequency components from local video features. Our approach effectively combines the consistency of global videos with the high fidelity of local videos, achieving both high fidelity and temporal consistency.

## 5.5 Multi-Prompt Video Generation

Our method can be seamlessly extended to multi-prompt video generation by providing different prompts for each video segment. As illustrated in Figure 7, our approach ensures coherent visual continuity and motion consistency. For instance, Ironman is shown running on the road, then standing, and finally flying into the sky, all within a consistent scene and with smooth action transitions. In the second row, we demonstrate a more complex prompt sequence describing weather and scene transitions. Our method effectively models the transition from "sunrise" to "storm with heavy rain and thunder" to the final "rainbow," maintaining consistency and capturing the fine-grained details of each prompt transition.

## 5.6 Longer Video Generation

To examine the scalability of our FreeLong, we extend the video generation length beyond 128 frames. As depicted in Figure 8, FreeLong effectively generates videos with even longer durations, such as

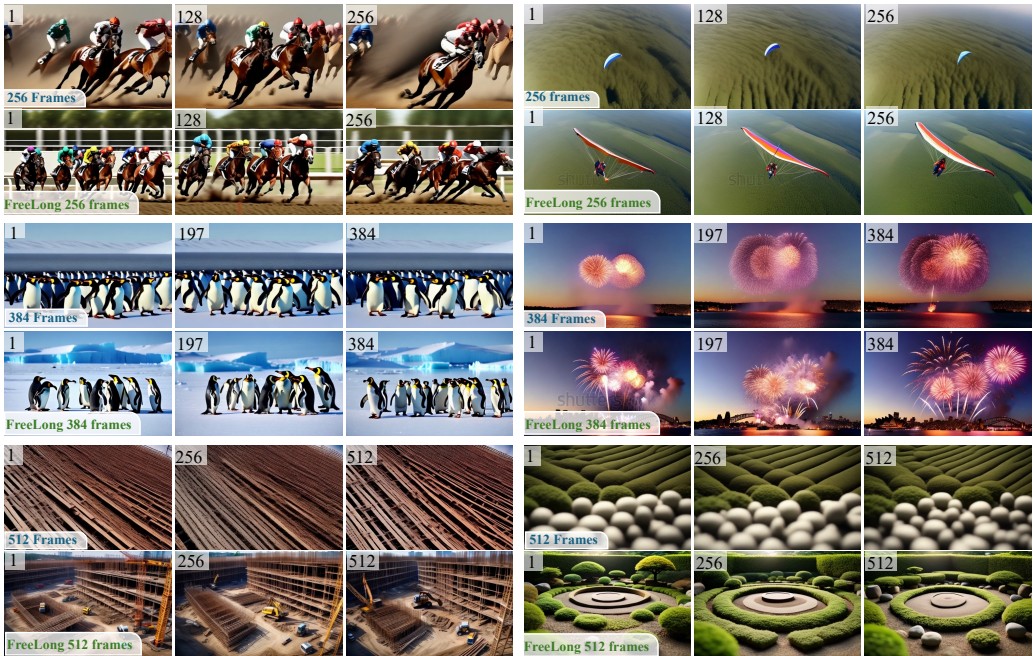

Figure 8: **Longer Video Generation.** FreeLong scales to generate videos longer than 128 frames (e.g., 512 frames), maintaining both temporal consistency and high fidelity across the entire sequence.

512 frames, while maintaining both temporal consistency and high fidelity throughout the entire sequence. This demonstrates that our method scales well with increasing video lengths, addressing the challenges associated with generating long continuous content without significant degradation in quality.

# 6 Conclusion

In this paper, we introduced FreeLong, a training-free method to adapt short video diffusion models for long video generation. Our research reveals that directly generating long videos from short video diffusion models results in poor quality, primarily due to high-frequency distortion. To resolve this issue, we employ the SpectralBlend Temporal Attention (SpectralBlend-TA) mechanism, which blends low-frequency global features with high-frequency local features to enhance consistency and fidelity in long videos. Our experiments demonstrate that FreeLong significantly outperforms existing models, achieving superior temporal consistency and video fidelity. Our experiments show that FreeLong significantly outperforms existing models, achieving better temporal consistency and video fidelity. FreeLong also supports coherent multi-prompt generation, offering a practical solution for high-quality long video creation without extensive retraining.

# 7 Acknowledgments

This work is supported by National Science and Technology Major Project (2022ZD0117802) and the National Natural Science Foundation of China (U2336212). This work is partially supported by the Fundamental Research Funds for the Central Universities (Grant Number: 226-2024-00058).

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

# A Appendix

## A.1 Social Impacts

It is important to consider the potential ethical implications of our approach, which is typical in generative models. By incorporating Video Diffusion Model [1, 2] into our methodology, there is a chance that our system may also inherit the biases present in these models. Additionally, we need to be aware of the potential risks, including the generation of deceptive, harmful, or discriminatory content.

## A.2 Limitation

Despite its significant advancements, FreeLong has several limitations. Temporal flickering can still occur in extended sequences, affecting the visual consistency over prolonged videos. Additionally, handling dynamic scene changes where context and content vary significantly remains challenging, as the current model may struggle to adapt to rapidly changing scenarios. Nonetheless, FreeLong represents a promising approach to training-free long-form text-to-video generation, offering significant improvements in consistency and fidelity despite these challenges.

## A.3 Code used and License

All used codes and their licenses are listed in Table 2.

Table 2: The used codes and license.

| URL | Citation | License |
| --- | --- | --- |
| https://github.com/Vchitect/LaVie | [1] | Apache License 2.0 |
| https://github.com/huggingface/diffusers | [47] | Apache License 2.0 |
| https://github.com/AILab-CVC/VideoCrafter | [2] | Apache License 2.0 |
| https://github.com/modelscope/modelscope | [5] | Apache License 2.0 |

## A.4 More Qualitative Results

We add more video generation results in Figure 9

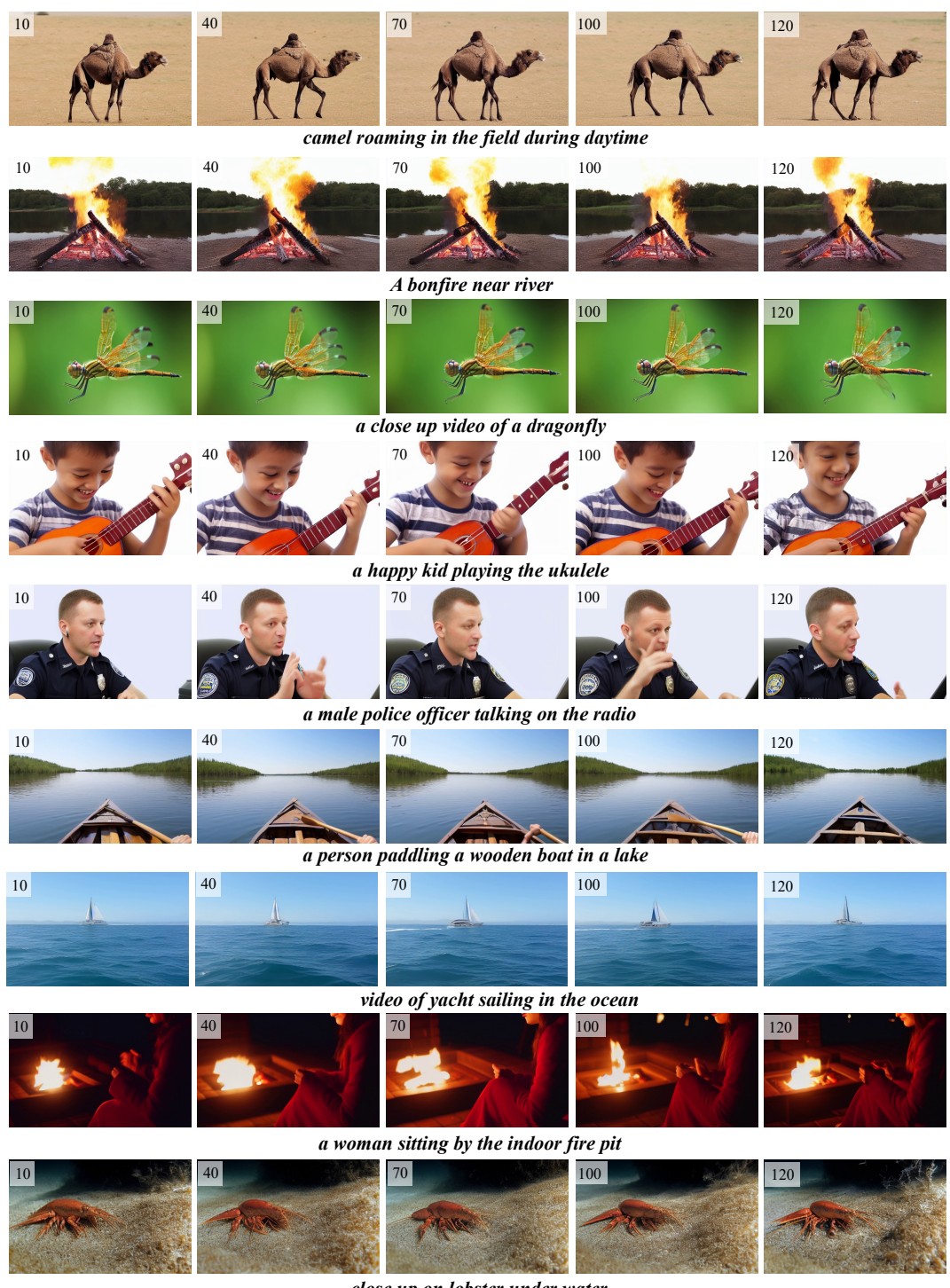

Figure 9: **More Long Video Generation Results.**

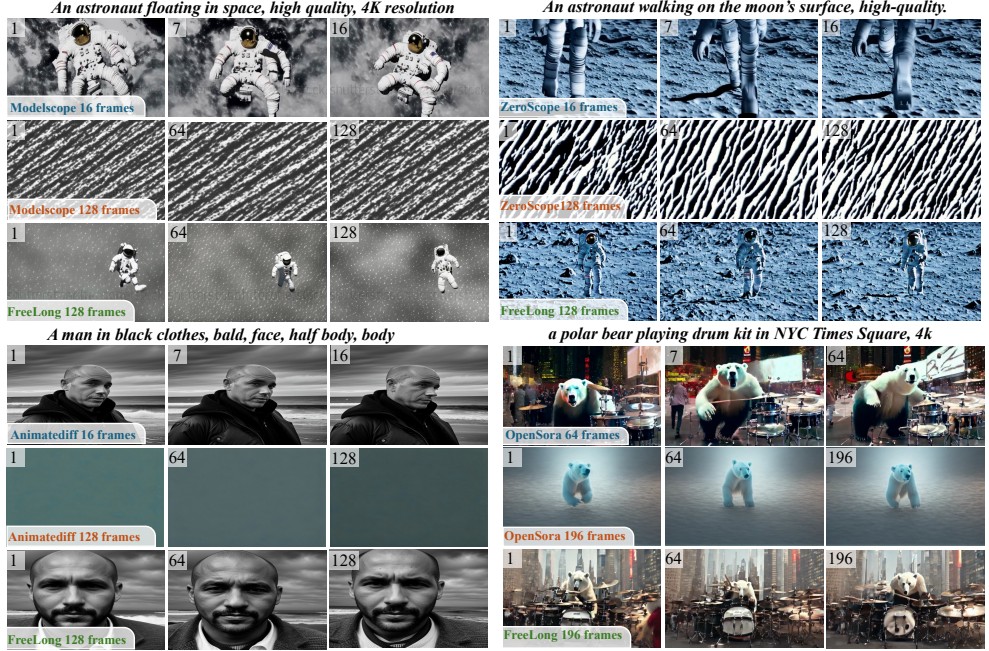

Figure 10: **Generalization to Other Base Models.** FreeLong can be easily integrated into various video diffusion frameworks by replacing the temporal attention with SpectralBlend-TA, enabling these models to generate consistent long videos with high fidelity.

## A.5 Generalization to Other Base Models

Our FreeLong method is designed to be model-agnostic and can be seamlessly integrated into various pre-trained short video diffusion models. To validate this, we apply FreeLong to other state-of-the-art video diffusion frameworks beyond LaVie [1] and VideoCrafter [2]. Specifically, we incorporate FreeLong into models like ModelScope [5], Animatediff [4] and OpenSora [48], replacing their original temporal attention mechanisms with our proposed SpectralBlend Temporal Attention. As illustrated in Figure 10, FreeLong successfully enhances these models' capabilities to generate consistent long videos with high fidelity. The generated videos maintain temporal coherence across extended sequences while preserving fine-grained spatial-temporal details. The successful integration and performance improvement across different models highlight the generalizability of our approach. This flexibility allows researchers and practitioners to extend the capabilities of various short video diffusion models without additional training.

