# OpenReview forum: "FreeLong: Training-Free Long Video Generation with SpectralBlend Temporal Attention"
_NeurIPS.cc/2024/Conference — NeurIPS 2024 poster_

### Official Review · Reviewer_QZ6p · 2024-07-08

**Soundness:** 3
**Presentation:** 2
**Contribution:** 3
**Rating:** 5
**Confidence:** 5

**Summary:**

The paper introduces a SpectralBlend Temporal Attention (SB-TA) mechanism, which blends low-frequency and high-frequency components from attention features together to enhance consistency and realism in generating long videos. The authors tested the proposed algorithm using 25 text prompts on LaVie and VideoCrafter, demonstrating improvements in VBench metrics. Additionally, they conducted ablation studies to showcase the characteristics of different modules in the proposed method.

**Strengths:**

1. The proposed method achieved notable improvements, and the authors conducted interesting analyses to support their findings.
2. The proposed method supports multi-prompt long video generation.

**Weaknesses:**

1. My first concern is that 128 frames itself cannot be considered long video generation. Therefore, it would be more convincing to see how effective the authors' proposed method is on longer videos, such as generating 1-minute videos like Sora.
2. The proposed method includes several hyperparameters, such as alpha and tau. Are these parameters significant in influencing the final results? Because the authors tested only 25 prompts, it's unclear whether these hyperparameters remain suitable across a broader range of scenarios.
3. Although adding local features to long-range features can mitigate some loss of spatial details,  for longer video generation with significant content variation, it remains uncertain whether these local features adequately supplement spatial details.

**Questions:**

1. Figure 2's right panel demonstrates strong diagonal correlations in VideoCrafter at 64-frame and 128-frame intervals. How does this correlate with the findings proposed by the authors?
2. Since the focus of this paper is to test the algorithm's ability in long video generation, while VBench primarily benchmarks the quality of short video generation, the key question is how to demonstrate that the results obtained reflect the algorithm's performance in long video generation quality.

**Limitations:**

The method proposed by the authors can only generate videos up to 128 frames, which falls short of true long video generation. It has yet to be validated in generating genuinely long videos.

---

> ### Author Rebuttal · Authors · 2024-08-07
>
> > **My first concern is that 128 frames itself cannot be considered long video generation. Therefore, it would be more convincing to see how effective the authors' proposed method is on longer videos, such as generating 1-minute videos like Sora.**
> >
>
> Thank you for your feedback regarding the length of the generated videos. Following other works such as [1,2,3], which consider videos longer than 10 seconds as "long," we chose 128 frames to speed up our experiments. We understand that 128 frames may not fully represent long video generation compared to examples like Sora. It is important to note that our FreeLong method is not limited to this length. To address your concern, we have conducted further experiments evaluating FreeLong on generating longer videos of 256, 384, and 512 frames (approximately 1 minute at 8 fps). These results, shown in Figure 1 of the submitted one-page PDF, indicate that FreeLong achieves robust longer video generation across these lengths.
>
> [1] StreamingT2V: Consistent, Dynamic, and Extendable Long Video Generation from Text
>
> [2] A Survey on Long Video Generation: Challenges, Methods, and Prospects
>
> [3] FreeNoise: Tuning-Free Longer Video Diffusion via Noise Rescheduling
>
> > **The proposed method includes several hyperparameters, such as alpha and tau. Are these parameters significant in influencing the final results?**
> >
>
> We appreciate your question regarding the significance of hyperparameters such as $\alpha$ and $\tau$. We set $\alpha$ to match the pre-trained video length. To explore the impact of $\tau$, we conducted quantitative ablation studies, as shown in the table below:
>
> | $\tau$ | sub consistency | back consistency | motion_smooth | imaging_quality |
> | --- | --- | --- | --- | --- |
> | 10 | 0.9236 | 0.9323 | 0.9685 | 0.6830 |
> | 20 | 0.9350 | 0.9358 | 0.9709 | 0.6893 |
> | 30 | 0.9462 | 0.9578 | 0.9758 | 0.6894 |
> | 40 | 0.9667 | 0.9660 | 0.9754 | 0.6913 |
> | 50 | 0.9601 | 0.9645 | 0.9736 | 0.6941 |
>
> Bigger $\tau$ means more steps to blend global and local features, the table shows that our method keeps enhancing the consistency when the steps increase and saturated at step 40.
>
> > **Because the authors tested only 25 prompts, it's unclear whether these hyperparameters remain suitable across a broader range of scenarios.**
> >
>
> Regarding the number of test prompts, as described in Lines 178-179, we randomly selected 25 prompts from each of the 8 test categories in VBench, resulting in a total of 200 prompts to ensure a fair comparison. This approach ensures a fair and comprehensive comparison across all categories. All test prompts are illustrated in the supplementary materials.
>
> > **Although adding local features to long-range features can mitigate some loss of spatial details, for longer video generation with significant content variation, it remains uncertain whether these local features adequately supplement spatial details.**
> >
>
> Thank you for pointing out the concern regarding the adequacy of spatial detail supplementation. We use the MUSIQ model, which evaluates over-exposure, noise, and blur of image, to validate image quality of the generated longer videos. The results are as follows:
>
> | Method | 16 frame | Direct | Sliding Window | FreeNoise | Ours |
> | --- | --- | --- | --- | --- | --- |
> | Imaging Quality | 0.6890 | 0.6298 | 0.6591 | 0.6645 | 0.6913 |
>
> Our method achieves comparable image quality to 16-frame videos, and significantly outperforms other methods. More qualitative examples on longer videos and additional base models are provided in Figure 1,2 in the one-page pdf.
>
> > **Figure 2's right panel demonstrates strong diagonal correlations in VideoCrafter at 64-frame and 128-frame intervals. How does this correlate with the findings proposed by the authors?**
> >
>
> Thank you for your valuable question. VideoCrafter2 is pre-trained on varied video lengths and is relatively robust to longer video generation than LaVie. To better understand the temporal relationships in temporal attention, we set the diagonal elements, such as (0,0), (1,1) to 0, which makes visualization of relationships between different frames more effective. As shown in Figure 4 of one-page pdf, the attention map reveals less structured attention patterns. This approach helps explain the long-range temporal attention of VideoCrafter2.
>
> > **how to demonstrate that the results obtained reflect the algorithm's performance in long video generation quality.**
> >
>
> Thank you for your valuable suggestions. VBench is a comprehensive evaluation benchmark for video diffusion models, with evaluation tools not constrained by video length, such as CLIP similarity, DINO similarity, and Temporal Flickering. Since no long video benchmark was available before the paper submission, we used this benchmark. Per your suggestion, we compared our method on the VBench-Long [4], designed for Sora-like video evaluation, as shown in the table below. It is evident that our method achieves consistent performance improvement compared to other methods.
>
> | Method | sub consistency | back consistency | motion_smooth | imaging_quality |
> | --- | --- | --- | --- | --- |
> | Direct | 0.8687 | 0.9151 | 0.9229 | 0.6298 |
> | Sliding Window | 0.8811 | 0.9089 | 0.9574 | 0.6591 |
> | FreeNoise | 0.9397 | 0.9511 | 0.9696 | 0.6645 |
> | FreeLong(Ours) | 0.9667 | 0.9660 | 0.9754 | 0.6913 |
>
> [4] https://github.com/Vchitect/VBench/tree/master/vbench2_beta_long

---

> > ### Comment · Reviewer_QZ6p · 2024-08-12
> >
> > Thank you for the authors' response. The rebuttal addresses most of my concerns. I would like to raise my score.

---

> > > ### Author Response · Authors · 2024-08-12
> > > **Thank you!**
> > >
> > > Thank you for your thoughtful review and for recognizing our efforts to address your concerns. We sincerely appreciate your consideration and insights. Thank you once again!

---

### Official Review · Reviewer_XZSx · 2024-07-10

**Soundness:** 3
**Presentation:** 2
**Contribution:** 2
**Rating:** 5
**Confidence:** 4

**Summary:**

This paper proposes a training-free method to generate 8x longer videos based on 16-frame pre-trained video diffusion models. It observes that extended temporal attention has a negative effect on high-frequency generation. Thus, it proposes an SB-TA module to fuse global low-frequency temporal features and local high-frequency temporal features.

**Strengths:**

1. Lines 42-43 & 107-121: The observation on SNR with spatial-temporal components is natural and reasonable.

2. Novelty: While frequency splitting and merging are common in low-level image processing tasks like super-resolution, this is the first solution for video temporal "super-resolution".

**Weaknesses:**

1. Lines 69-76: The related work section is not reader-friendly for those unfamiliar with the field. Simply listing papers across a wide range (from GAN to diffusion) is too vague to introduce the outline of the research field.

2. Lack of comparisons and baselines: The observation analysis and experiments only include two base models: LaVie and VideoCrafter. Note that both of these video diffusion models adopt relative positional encoding (PE) in their temporal attention blocks. Other models like AnimateDiff (absolute PE), DIT-based models like Open-Sora (RoPE), and ModelScope (which adds temporal convolution, no PE) are not discussed in the paper. This suggests that the proposed method may be limited to one specific PE.

3. The visual result quality is limited and does not include more motions, despite the frame rate being 8x (16 to 128). Although the authors show multi-prompt generation ability in Sec. 5.5, a single prompt should still be enough to cover a much longer motion. (e.g., Fig.7's first prompt, "running Iron Man," is a complex motion, and it should be generated correctly as a complete motion with 128 frames sampling. Can you show this case in the rebuttal?)

**Questions:**

1. In Lines 33-38 and Fig. 1, the authors claim that longer video generation on short-clip pre-trained models will degrade high-frequency details. However, the results of FreeLong still do not have rich details (especially in the background).

2. Did you consider the memory cost of long-video generation with one forward pass (128 frames)?

**Limitations:**

Yes.

---

> ### Author Rebuttal · Authors · 2024-08-07
>
> Thank you for your valuable feedback. We have carefully considered your comments and suggestions to improve our paper. Below are our responses to each of your concerns:
>
> > **The related work section is not reader-friendly for those unfamiliar with the field**
> >
>
> We apologize for the lack of clarity in the related work section. We understand that it may be challenging for readers unfamiliar with the field. To address this, we will revise this section to provide a more comprehensive overview of the research landscape. We will include a structured summary that highlights the key developments and contributions in video diffusion models, making it more accessible and informative for all readers.
>
> > **Lack of comparisons and baselines**
> >
>
> We appreciate your insightful feedback regarding the need for more comparisons and baselines. Initially, we chose LaVie and VideoCrafter2 due to their state-of-the-art performance in video generation. However, we recognize the importance of demonstrating the generalizability of our proposed method, FreeLong, across different base models.
>
> In response, we have extended our experiments to include additional models, such as OpenSORA, AnimateDiff, and ModelScope. Figure 2 in the one-page pdf presents these results, showing that FreeLong performs well across these diverse models. This broader evaluation demonstrates that FreeLong is not limited to a specific positional encoding and can be effectively applied to various video diffusion models.
>
> > **The visual result quality is limited and does not include more motions**
> >
>
> We understand your concern about the visual result quality and the limited motion content in our generated videos. For single-prompt inputs, we focus on maintaining consistency a cross long sequences, while for multi-prompt inputs, we aim to facilitate more complex and accurate motions.
>
> As illustrated in Figure 3 of the one-page pdf, our method achieves more natural variations compared to FreeNoise. FreeNoise often generates repetitive content due to repeated noise initialization to maintain temporal consistency. In contrast, FreeLong utilizes global temporal attention to explicitly maintain temporal consistency without sacrificing motion variety.
>
> > **FreeLong still do not have rich details**
> >
>
> Thank you for pointing out the need for richer high-frequency details, particularly in the background. FreeLong aims to improve both consistency and fidelity in long video generation. To validate spatial details of generated videos, we used the MUSIQ model from VBench to evaluate image quality~(e.g. over-exposure, noise, blur). The results are shown in the table below:
>
> | Method | 16 frame |  Direct | Sliding Window | FreeNoise | Ours |
> | --- | --- | --- | --- | --- | --- |
> | Imaging Quality | 0.6890 | 0.6298 | 0.6591 | 0.6645 | 0.6913 |
>
> Our method achieves comparable image quality to 16-frame videos, demonstrating improved spatial detail. More qualitative examples on longer videos and additional base models are provided in Figure 1,2 in the one-page pdf.
>
> > **Memory Cost**
> >
>
> We have calculated the computational cost, including inference time and memory usage, as shown in the table below:
>
> | Method | Inference Time | Memory Cost |
> | --- | --- | --- |
> | Direct | 1.8s | 18251MiB |
> | Sliding Window | 2.6s | 15017MiB |
> | FreeNoise | 2.6s | 15017MiB |
> | FreeLong(Ours) | 2.2s | 20179MiB |
>
> Compared to the direct application method, our FreeLong method slightly increases both memory cost and inference time to achieve consistent and high-fidelity long video generation. FreeLong can generate 128-frame videos using a single NVIDIA RTX 4090 GPU, ensuring feasibility for practical use.
> When compared to FreeNoise, FreeLong reduces the inference time per step from 2.6s to 2.2s while increasing memory usage. This highlights FreeLong's efficiency in generating longer videos, as it effectively balances inference time with memory cost.
> By optimizing these trade-offs, FreeLong demonstrates its capability to generate longer videos efficiently.

---

> > ### Comment · Reviewer_XZSx · 2024-08-09
> >
> > Thank you for the comprehensive response from the authors, which has addressed most of my concerns; I believe the authors need to highlight the expansion on other models in subsequent versions; my only remaining concern is regarding the effectiveness, as I think the video results of long sequence generation shown in the PDF still do not fully demonstrate that FreeLong can effectively generate complete complex motions. I will raise the evaluation to 5.

---

> ### Author Response · Authors · 2024-08-11
> **Thank you！**
>
> Thank you for your thorough review and constructive feedback. We greatly appreciate your thoughtful comments and are pleased that our response has addressed most of your concerns. We will highlight the expansion on other models in future versions. Regarding the demonstration of FreeLong's effectiveness in generating complex motions over longer sequences:
>
> Due to the 50 MB limit of the one-page PDF, we could only provide a few images for each example. While these images demonstrate the generalizability of our approach, they may not fully capture the complexity of the motions FreeLong can produce. Videos offer more coherence and motion variation than images. We would like to highlight that, compared to previous methods like FreeNoise, FreeLong achieves consistent long video generation with more natural motions, as shown in Figure 3 of the one-page PDF.
>
> In future versions, we will include more detailed examples to showcase FreeLong's capability to handle complex motions in longer sequences effectively.
>
> Thank you once again for your valuable insights, which have been instrumental in enhancing our work.

---

### Official Review · Reviewer_oAuU · 2024-07-10

**Soundness:** 4
**Presentation:** 4
**Contribution:** 3
**Rating:** 6
**Confidence:** 3

**Summary:**

The auther propose FreeLong, a training-free method for long video generation. This paper identifies that the problem with long video generation lies in the scope of attention and the distortion of high-frequency components. Based on the observation, the auther proposes a novel method to blend different frequency components of global and local video, leading to better result in long video generation.

**Strengths:**

1. The research problem is important. The paper is well-written and easy to follow.
2. The observation and the method are reasonable.
3. Experiments results look promising.

**Weaknesses:**

My main concern lies in the computational cost.

**Questions:**

1) Computing the global attention map and video features seems computationally expensive, what is the computational cost compared to previous method (like freenoise) ?
2) Method like freenoise suffer from repetitive generation. Generated contents will repeat several times. Can freelong solve this problem?

**Limitations:**

Yes

---

> ### Author Rebuttal · Authors · 2024-08-07
>
> > **what is the computational cost compared to previous method (like freenoise) ?**
> >
>
> Based on your feedback, we conducted a comparison of inference times between our method, FreeLong, and other methods, including FreeNoise, the sliding window method, and direct application. The results are summarized in the table below:
>
> | Method | Inference Time |
> | --- | --- |
> | Direct | 1.8s |
> | Sliding Window | 2.6s |
> | FreeNoise | 2.6s |
> | FreeLong(Ours) | 2.2s |
>
> Our method, FreeLong, utilizes both global and local attention streams, performing a single-pass temporal attention forward operation. In contrast, previous methods like FreeNoise rely on sliding window temporal attentions, which require multiple rounds of temporal attention and thus increase inference time. Our results demonstrate that FreeLong achieves faster inference time.
>
> > **Can freelong solve the repetitive generation problem?**
> >
>
> Regarding the repetitive generation problem observed in method FreeNoise, it is primarily due to their reliance on repetitive noise initialization. Our proposed method, FreeLong, overcomes this issue by explicitly capturing global consistency during the temporal attention process. This enables FreeLong to generate consistent longer videos without depending on repetitive initial noises.
>
> By effectively capturing global consistency, FreeLong produces more diverse and coherent video sequences, which significantly reduces the repetitive generation problem. Figure 3 in the submitted one-page PDF provides a comparison of the results between FreeLong and FreeNoise, demonstrating our approach's effectiveness in reducing repetitive generation while maintaining high-quality video generation results.

---

> > ### Comment · Reviewer_oAuU · 2024-08-09
> >
> > Thank authors for the response. You have addressed my concerns.

---

> > > ### Author Response · Authors · 2024-08-11
> > > **Thank you!**
> > >
> > > Thank you for your thoughtful review and for recognizing our efforts to address your concerns. We are pleased to hear that you found our explanations on computational efficiency and reducing repetitive generation helpful. We sincerely appreciate your consideration and insights. Thank you once again!

---

### Official Review · Reviewer_838f · 2024-07-13

**Soundness:** 3
**Presentation:** 3
**Contribution:** 3
**Rating:** 7
**Confidence:** 4

**Summary:**

The paper presents FreeLong, a novel training-free method for generating extended videos (128 frames) using pre-trained short video (16 frames) diffusion models. The key component is the SpectralBlend Temporal Attention (SB-TA), which fuses low-frequency global video features with high-frequency local features to ensure consistency and fidelity in long video generation. Experimental results highlight FreeLong's superior performance in video fidelity and temporal consistency over existing methods.

**Strengths:**

1. **Comprehensive Analysis**: The paper conducts an in-depth frequency analysis to identify challenges in long video generation and supports the proposed solution with extensive experimental validation.
2. **Novel SB-TA Mechanism**: The proposed SpectralBlend Temporal Attention effectively mitigates high-frequency component degradation, ensuring consistent and high-fidelity video outputs.
3. **Training-Free Long Video Generation**: Long video generation is an important task and future direction. I am happy to see a training-free method appear in the community. The proposed FreeLong is both resource-efficient and practical, allowing for the adaptation of existing short video models to long video generation without the need for retraining.
4. **Multi-Prompt Generation Support**: The multi-prompt video generation is coherent within FreeLong, and the visual continuity and smooth transitions between different scenes are satisfactory.

**Weaknesses:**

1. Inference time comparisons should include both single-pass and multi-pass temporal attention from previous methods to demonstrate the advantages clearly.
2. Although the video generation results are impressive, a more extensive user study is required to validate the consistency and quality of the generated videos.

**Questions:**

Some concerns please refer to the Weaknesses.
Also what about the longer duration of video generation, for example, videos longer than 128 frames?

**Limitations:**

Including examples of failure cases would provide a better understanding of the method's limitations.

---

> ### Author Rebuttal · Authors · 2024-08-07
>
> > **Inference time comparisons**
> >
>
> Thank you for your question. Following your advice, we conducted a comparison of inference times. As shown in the table below, our method achieves faster inference speeds than multi-pass methods, such as FreeNoise.
>
> | Method | Inference Time |
> | --- | --- |
> | Direct | 1.8s |
> | Sliding Window | 2.6s |
> | FreeNoise | 2.6s |
> | FreeLong(Ours) | 2.2s |
>
> > **User study**
> >
>
> Thank you for your valuable suggestions. We conducted a user study to evaluate temporal consistency, fidelity, and overall rankings. Ten participants from academia and industry took part in this study. The results are as follows:
>
> | Method | Consistency $\uparrow$ | Fidelity $\uparrow$ | Overall rank $\downarrow$ |
> | --- | --- | --- | --- |
> | Direct | 2.47 | 1.85 | 3.96 |
> | Sliding Window | 2.03 | 2.30 | 3.45 |
> | FreeNoise | 2.36 | 2.81 | 1.48 |
> | FreeLong(Ours) | 3.14 | 3.04 | 1.11 |
>
> Our method, FreeLong, outperformed other methods across all evaluated criteria.
>
> > **Longer Videos**
> >
>
> We have provided longer videos, including results with 256, 384, and 512 frames, in the one-page PDF. These results demonstrate the generalization ability of FreeLong to handle longer videos.

---

> > ### Comment · Reviewer_838f · 2024-08-09
> > **Good rebuttal**
> >
> > Thank authors for the thorough response and the additional experiments results. I am satisfied with the rebuttal and no longer have any further concerns. Therefore, I have decided to raise my score to accept.

---

> > > ### Author Response · Authors · 2024-08-09
> > > **Thank you!**
> > >
> > > We greatly appreciate the reviewer's prompt response and thoughtful evaluation of our work! Your positive feedback and constructive comments are truly valuable in helping us refine and improve our project. We would like to express our heartfelt gratitude for your time and consideration. Thank you once again!

---

### Author Rebuttal · Authors · 2024-08-07

We thank all reviewers for engaging in the review process. Our code will be made public upon acceptance.

We are deeply encouraged by positive comments from the reviewers. We appreciate the recognition and endorsement of our proposed training-free pipeline, such as acknowledging its analysis and method as reasonable (**838f,oAuU**), interesting (**QZ6p**), novel (**XZSx**), and effective (**838f,**). **838f**, **oAuU**, and **QZ6p** agree that our method generates videos that achieved notable improvements.

In our individual replies, we attempted to address specific questions and comments as clearly and in detail as possible.

Moreover, we added several additional results to the one-page PDF and the individual replies to strengthen our work overall. Here, we briefly summarize these additional experiments and evaluations:

- Longer videos from 256 to 512 frames
- Generalization to other base models with different position encoding, including ModelScope, Animatediff, and OpenSORA
- Detail comparison to FreeNoise
- More clear attention visualization on VideoCrafter2

We hope that these additional results further strengthen FreeLong position as the state-of-the-art method of training-free extending short video diffusion models to generate longer sequences and demonstrate:

- Our FreeLong can effectively generate longer video sequences not limited by fixed 128 frames.
- Our FreeLong is robust and is compatible to different video diffusion models and enhances the long context consistency and fidelity.

---

### Decision · Program_Chairs · 2024-09-25

**Decision:**

Accept (poster)

**Comment:**

This paper received four positive ratings after the rebuttal. The reviewers are satisfied with the rebuttal since it addressed the unclear issues in method design and experimental performance. The AC has read the paper, reviews, and rebuttal, making the decision of acceptance based on the consensual opinion of the reviewers. However, the authors still need to address the remaining issues regarding the approach effectiveness as suggested by the reviewers.